# Electrothermal Monte Carlo Simulation of a GaAs Resonant Tunneling Diode

**Orazio Muscato** 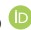

Department of Mathematics and Computer Science, University of Catania, 95125 Catania, Italy; orazio.muscato@unict.it

**Abstract:** This paper deals with the electron transport and heat generation in a Resonant Tunneling Diode semiconductor device. A new electrothermal Monte Carlo method is introduced. The method couples a Monte Carlo solver of the Boltzmann–Wigner transport equation with a steady-state solution of the heat diffusion equation. This methodology provides an accurate microscopic description of the spatial distribution of self-heating and its effect on the detailed nonequilibrium carrier dynamics.

**Keywords:** Monte Carlo methods; statistical mechanics of semiconductors; heat transfer

**MSC:** 65C05; 82D37; 80M31

## 1. Introduction

Due to the continued miniaturization of integrated circuits and the current trend toward nanoscale electronics, power densities, heat generation, and chip temperatures will reach levels that will prevent the reliable operation of such circuits. In order to minimize self-heating effects, the development of accurate electrothermal simulators is required, which takes into account the coupling between electronic and lattice dynamics. In the active regions of such small devices, heat generation is a direct consequence of the nonequilibrium carrier transport. In high electric field regions, the electrons are accelerated and collide with the lattice in such a way that the emission of a large number of phonons contributes to heat transport in the device. In the framework of semiclassical charge transport, electrothermal simulators are based on drift–diffusion or hydrodynamic models [1,2], which are able to capture nonequilibrium transport effects. Alternatively, the direct simulation Monte Carlo (MC) can provide an accurate nonequilibrium charge transport simulation, which is free from the approximations made in the drift–diffusion or hydrodynamic model. Electrothermal Monte Carlo simulators have been developed during these years [3–5] but not in quantum regimes where the Boltzmann Transport Equation must be replaced by the Wigner Transport Equation (WTE). Since electron devices are quantum systems outside of thermodynamic equilibrium, scattering by phonons should be included in the WTE for a realistic simulation. Many proposals for the collision operator can be found in the literature [6,7], which provide an accurate description of the phenomena at the price of a high requirement of computational resources. Because of that, the use of such operators is restricted to very simple (idealized) systems. In this paper, the effects of scattering with phonons are taken into account via a semiclassical Boltzmann collision operator, which employs transition rates calculated using Fermi's golden rule, obtaining the so called Boltzmann–Wigner transport equation (BWTE). Numerical solvers of the WTE can be based on finite-difference schemes [8–13], where scattering was restricted to the relaxation time approximation and the momentum space to one dimension. The Monte Carlo method allows for scattering processes to be included on a more detailed level, assuming a three-dimensional momentum–space. In this paper, we shall use the so-called *Signed Particle Monte Carlo* method (SPMC) [14,15] in which the effect of the Wigner potential is interpreted as a probabilistic generation of couples of positive and

negative particles, where the quantum information is carried by their sign. The huge number of generated particles can be controlled by an annihilation process: two particles with an opposite sign entering a given phase space cell are canceled. Recently, this method has also been understood in terms of the Markov jump process theory [16], producing a class of new stochastic algorithms. Algorithms that belong to this class are a standard time-splitting algorithm and a new no-splitting algorithm that avoids errors due to time-discretization [17,18].

Taking advantage of previous Electrothermal Monte Carlo semiclassical models, in this paper, we shall study the heating effect in a Resonant Tunneling Diode (RTD), coupling the SPMC solver of the BWTE with a steady-state solution of the heat diffusion equation. To the author's knowledge, this model is the first of its kind in terms of model accuracy. The paper is organized as follows. Details of the Boltzmann–Wigner transport equation are provided in Section 2, and in Section 3 we deal with the *Signed Particle Monte Carlo* method. In Section 3, we introduce the Resonant Tunneling diode structure and in Section 5 the Electrothermal Signed Particle Monte Carlo Method. Simulation results are shown in Section 6, and conclusions are drawn in Section 7.

## 2. The Boltzmann–Wigner Transport Equation

The BWTE writes [19]

$$\frac{\partial}{\partial t} f_w(t, x, k) + \frac{\hbar}{m^*} k \cdot \nabla_x f_w(t, x, k) + \frac{e}{\hbar} \nabla_x \varphi \cdot \nabla_k f_w(t, x, k) = \mathcal{Q}(f_w) + \mathcal{C}(f_w). \tag{1}$$

$x \in \mathbb{R}^3$ and $\hbar k \in \mathbb{R}^3$ are the electron position and momentum, respectively, $m^*$ is the electron effective mass, and $\varphi$ the slowly-varying potential satisfying the Poisson equation

$$\nabla \cdot [\varepsilon_0 \varepsilon_r \nabla \varphi(x)] = -e(N_D - N_A - n), \tag{2}$$

where $e$ is the elementary charge, $\varepsilon_0$ the absolute dielectric constant, $\varepsilon_r$ the relative dielectric constant, $N_D$, $N_A$ are the donors and acceptors' doping profiles, and $n$ the particle density

$$n(t, x) = \int f_w(x, k, t) \, dk \quad . \tag{3}$$

$\mathcal{C}(f_w)$ is the Boltzmann scattering operator which, in the not-degenerate case, is as follows [20]:

$$\mathcal{C}(f_w) = \int \left[ w_s(k', k) f_w(k') - w_s(k, k') f_w(k) \right] dk', \tag{4}$$

where $w_s(k, k')$ is the scattering rate at which electrons suffer with phonons and impurities, given by the Fermi's golden rule. The quantum evolution is taken into account by the term

$$\mathcal{Q}(f_w) = \int_{\mathbb{R}^d} V_w(x, k - k') f_w(t, x, k') \, dk', \tag{5}$$

where $V_w$ is the Wigner potential

$$V_w(x, k) = \frac{1}{i\hbar(2\pi)^d} \int_{\mathbb{R}^d} dx' \, e^{-ik \cdot x'} \left[ V\left( x + \frac{x'}{2} \right) - V\left( x - \frac{x'}{2} \right) \right], \tag{6}$$

and $V$ is the rapidly-varying term of the potential energy.

## 3. The Signed Particle Monte Carlo Method

The quantum evolution term (5) can be interpreted like the *Gain* term of the collisional operator of the Boltzmann transport equation, in which the *Loss* term is missing. However, the Wigner potential (6) is not always positive and, for this reason, cannot be considered a

scattering term. The main idea of the *Signed Particle Monte Carlo* method [14] consists of separating $V_w$ into a positive and negative part $V_w^+, V_w^-$ such that

$$V_w = V_w^+ - V_w^- \quad , \quad V_w^+, V_w^- \geq 0 \tag{7}$$

Consequently, we can define an integrated scattering probability per unit time as

$$\gamma(x) = \int dk'\, V_w^+(x, k - k') = \int dk'\, V_w^-(x, k - k') \tag{8}$$

and rewrite the quantum evolution term as the difference between *Gain* and *Loss* terms, i.e.,

$$\mathcal{Q}(f_w) \;=\; \int dk'\, w(k', k) f_w(t, x, k') - \gamma(x) f_w(t, x, k) \tag{9}$$

$$w(k', k) \;=\; V_w^+(x, k - k') - V_w^-(x, k - k') + \gamma(x)\delta(k - k') \quad . \tag{10}$$

The interpretation of the scattering term $w(k', k)$ is that a particle produces, in the same position, a couple of new particles with weight $u$ and $-u$ according to a generation rate given by the function $\gamma(x)$. The momentum of the new particles is generated with probability $V_w^+(x, k)/\gamma(x)$. Since usually $\gamma$ is rapidly oscillating, an exponential growth of particle numbers is expected and, in order to control the particle number, a cancellation procedure is mandatory.

This procedure has been understood using the theory of the piecewise deterministic Markov processes [16], where the state space is

$$z_j(t) = (u_j(t), x_j(t), k_j(t)), \quad t \geq 0 \quad , \quad j = 1, \ldots, N(t), \tag{11}$$

and $u_j \in \{-1, +1\}$ is the weight. The time evolution of the particle system (11) is assigned by a deterministic motion according to the flow

$$F(t, z) = (u, x + v(k)t, k) \quad , \quad v = \frac{\hbar}{m} k \tag{12}$$

and a jump kernel $Q(z_j(t))$. The random waiting time $\tau$ until the next jump satisfies

$$\mathbb{P}(\tau \geq t) = \exp\left(-\int_0^t Q(F(s, z))\, ds\right) \quad . \tag{13}$$

For numerical purposes, we introduce a majorant $\hat{V}_w$ such that

$$|V_w(x, k)| \leq \hat{V}_w(x, k) \quad \forall x, k \in \mathbb{R}^3 \quad . \tag{14}$$

If the $j$-th particle generates two new particles with

$$z_1' = (u_j\, \mathrm{sign}\, V_w(x_j, k), x_j, k_j + k) \quad , \quad z_2' = (u_j\, \mathrm{sign}\, V_w(x_j, k), x_j, k_j - k) \tag{15}$$

the jump kernel takes the form [17]

$$Q(z_j) = \frac{1}{2} \int \hat{V}_w(x_j, k)\, dk \tag{16}$$

and Equation (13) writes

$$\mathbb{P}(\tau \geq t) = \exp\left(-\int_0^t \hat{\gamma}(x_j + v(k_j)s)\, ds\right) \tag{17}$$

where

$$\hat{\gamma}(x) = \frac{1}{2} \int \hat{V}_w(x, k)\, dk \tag{18}$$

represents the generation probability. It is possible to prove that functionals of the solution of the Wigner equation are expressed in terms of the particle system using the representation [16]

$$\int \int \phi(x,k) f(t,x,k) dk dx = \frac{1}{N_{ini}} \mathbb{E}\left(\sum_{j=1}^{N(t)} u_j(t)\phi(x_j(t),k_j(t))\right) \qquad (19)$$

where $\phi$ is an appropriate test function, and $N_{ini} = N(0)$ is the initial particle number. In order to to separate the transport and the jump processes, usually a splitting time step $\Delta t$ is used at the expense of a discretization error. This can be avoided by using a no-splitting algorithm recently introduced in [17,18]. By introducing a majorant for the generation process (8) and one for the total phonon scattering rate

$$\Gamma_s \geq \max \lambda(k) \quad , \quad \lambda(k) = \sum_{\alpha} \int w_{\alpha}(k,k') \, dk' \qquad (20)$$

The total majorant is

$$\Gamma = \Gamma_s + \hat{\gamma} \qquad (21)$$

and Equation (17), for all particles, now is as follows:

$$\mathbb{P}(\tau \geq t) = \exp\left(-\sum_{j=1}^{N} \int_0^t \left[\Gamma_s + \hat{\gamma}(x_j + v(k_j)s)\right] ds\right) \quad . \qquad (22)$$

In the case in which $\hat{\gamma}$ does not depend on the position, we have

$$\mathbb{P}(\tau \geq t) = \exp(-\Gamma N t) \rightarrow \quad \tau = -\frac{1}{\Gamma N} \log r \qquad (23)$$

where $r \in U[0,1]$, and $\tau$ is completely determined. With respect to the splitting case, now the transport and the generation process can not be separated, and the results shall not be affected by any discretization error.

## 4. The Resonant Tunneling Diode

A standard Resonant Tunneling Diode structure [21] has been implemented, as shown in Figure 1. The barriers have depth $b = 3$ nm, height $a = 0.3$, and the quantum well dimension is $b_w = 5$ nm, symmetric with respect to the mid-point $L/2$ (total length $L = 150$ nm). The barrier structure is embedded in a 30 nm lightly doped region ($N_D = 10^{16}$ cm$^{-3}$) which is connected to 60 nm highly doped regions on either side ($N_D^+ = 10^{18}$ cm$^{-3}$).

In this case, the Wigner potential (6) can be easily evaluated in addition to the majorant (18) (see [18] for the details). The device considered is made by Gallium Arsenide (GaAs) (with $m^* = 0.067$), and polar optical phonons (POP) within a single $\Gamma$ band [20] in the parabolic band approximation used are taken into account. The total scattering rate is written as follows [20]:

$$\lambda(k,T_L) = \lambda^-(k,T_L) + \lambda^+(k,T_L) \qquad (24)$$

where the first term represents POP absorption and the second one emission

$$\lambda^-(k,T_L) = \frac{e^2 \omega_p \left(\frac{1}{\epsilon_\infty} - \frac{1}{\epsilon_r}\right)}{2\pi\epsilon_0 \hbar \sqrt{\frac{2\varepsilon(k)}{m^*}}} n_0 \sinh^{-1} \sqrt{\frac{\varepsilon(k)}{\hbar\omega_p}} \qquad (25)$$

$$\lambda^+(k,T_L) = \frac{e^2 \omega_p \left(\frac{1}{\epsilon_\infty} - \frac{1}{\epsilon_r}\right)}{2\pi\epsilon_0 \hbar \sqrt{\frac{2\varepsilon(k)}{m^*}}} (n_0 + 1) \sinh^{-1} \sqrt{\frac{\varepsilon(k)}{\hbar\omega_p} - 1} \qquad (26)$$

The term $n_0(T_L)$ is the phonon equilibrium distribution, i.e.,

$$n_0(T_L) = \frac{1}{\exp\left(\frac{\hbar\omega_p}{k_B T_L}\right) - 1} \tag{27}$$

$\hbar\omega_p$ is the polar optical phonon energy (0.03536 eV) and $T_L$ the lattice temperature. The initial lattice temperature is 300 K, and ohmic boundary conditions are used.

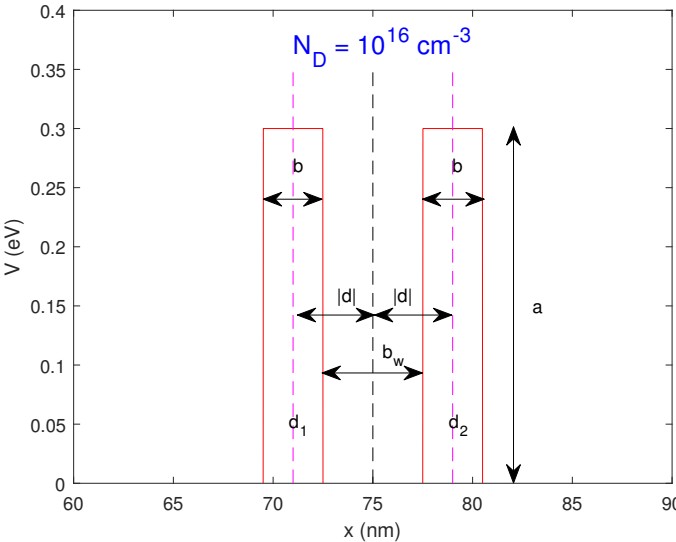

**Figure 1.** The quantum well region.

## 5. The Electrothermal Signed Particle Monte Carlo Method

An important issue that arises from the coupling of an MC electronic transport algorithm to any thermal model is the significant difference in the characteristic time scales of electronic and thermal transport. Electronic transients in GaAs systems are of the order of picoseconds, whereas thermal transients may be of the order of nanoseconds, microseconds, or even longer. Performing MC computations for the duration of thermal transients across the whole semiconductor die would not be feasible. Consequently, the method used in this paper extracts steady-state electrothermal device characteristics only. The electrothermal SPMC method of simulation is an iterative approach:

- The initial SPMC iteration is run at a room temperature of 300 K for a few ps, in order to reach a steady-state;
- As the steady state is reached, electronic parameters are sampled for typically 15 ps, in order to evaluate the heat generation rate $H(x)$;
- The lattice temperature $T_L(x)$ is obtained by solving the steady-state heat diffusion equation

$$\nabla_x(\kappa \nabla_x T_L(x)) + H(x) = 0 \tag{28}$$

  $\kappa$ being the thermal conductivity in GaAs;
- The SPMC solver is rerun, in the next iteration, with the new lattice temperature $T_L(x)$. We observe that the scattering rates (25) and (26) depend on the lattice temperature;
- We repeat this procedure until convergence is reached.

This model does not account for temperature changes beyond the semiconductor die. Radiation losses are neglected, as their contribution at the small die surface areas is insignificant.

The mechanism through which Joule heating occurs is that of electron scattering with phonons, and consequently only a simulation approach which deliberately incorporates all such scattering events will capture the complete microscopic, detailed picture of lattice heating. The phonon emission and absorption events during a simulation run are tallied and full heat generation statistics can be collected. We wait until the steady state has been reached at time $t_0$. Then, we count our events in the observation points $t_i$, $i = 0, \ldots, N_{obs}$. We evaluate the heat generation rate in two ways:

1. **Counting the phonon number.**
   We introduce the quantity [22]

$$H^c(t_{i-1}, t_i, x) = \frac{n(t_i, x)}{N_p(t_i, x)} \frac{\hbar\omega_p[C^+ - C^-]}{dt}, \tag{29}$$

where $C^+(t_{i-1}, t_i, x)$, $C^-(t_{i-1}, t_i, x)$ are the numbers of the phonon emitted and absorbed in the time interval $(t_{i-1}, t_i)$ in the $x$-th grid point, $n(t, x)$ the charge density, and $N_p(t, x)$ the particle number at time $t$ in the $x$-th cell. Then, the heat generation rate is

$$\langle H^c(x) \rangle = \frac{1}{N_{obs}} \sum_{i=1}^{N_{obs}} H^c(t_{i-1}, t_i, x) \tag{30}$$

2. using the **integrated probability scattering function**.
   From the integrated probability scattering (25) and (26) we can define

$$H^F(t_i, x) = \frac{n(t_i, x)}{N_{ini}} \sum_{j=1}^{N(t_i)} u_j G(\varepsilon(k_j)) \quad , \quad G(\varepsilon) = \hbar\omega_p[\lambda^+(k) - \lambda^-(k)] \tag{31}$$

Then, the heat generation rate is

$$\left\langle H^F(x) \right\rangle = \frac{1}{N_{obs}} \sum_{i=1}^{N_{obs}} H^F(t_i, x) \quad . \tag{32}$$

The heat generation is reduced to the usual calculation of functionals according to Equation (19). This estimator enjoys better approximation properties due to reduced statistical fluctuations [5].

## 6. Numerical Results

In order to have a significant lattice temperature increase with respect to the equilibrium temperature of 300 K, the applied bias voltage $V_b$ must be greater than 0.8 V. In Figure 2, we plot the heat generation rate versus the position, evaluated by means of the counting estimator (30) and the integrated probability estimator (32), for $V_b = 0.8$ V. From this figure, we can see that the maximum heat is produced inside the quantum well region, representing a so-called *hot spot region*.

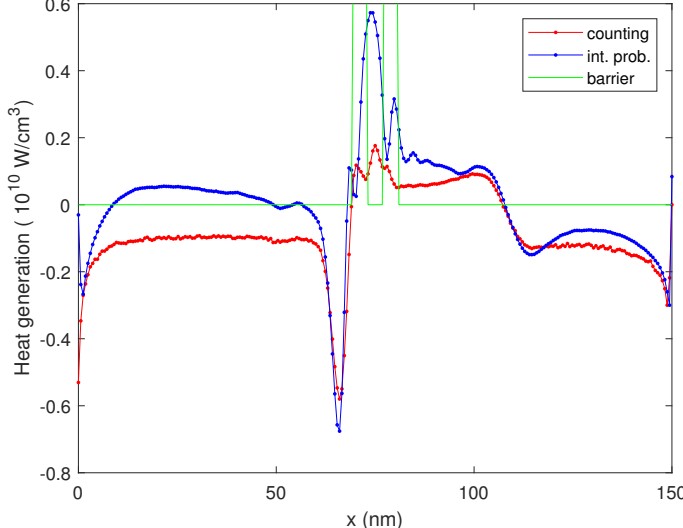

**Figure 2.** The heat generation rate versus the position for $V_b = 0.8$ V evaluated by means of the counting estimator (30) and the integrated probability estimator (32).

In Figure 3, we plot the corresponding standard deviation, proving the variance reduction of the integrated probability estimator (32). In Figure 4, we plot a zoom of Figure 2 with the error bar, proving that the integrated probability estimator is always inside the tolerance band of the counting estimator. Figure 5 shows the density for the first two iterations, showing no appreciable variation.

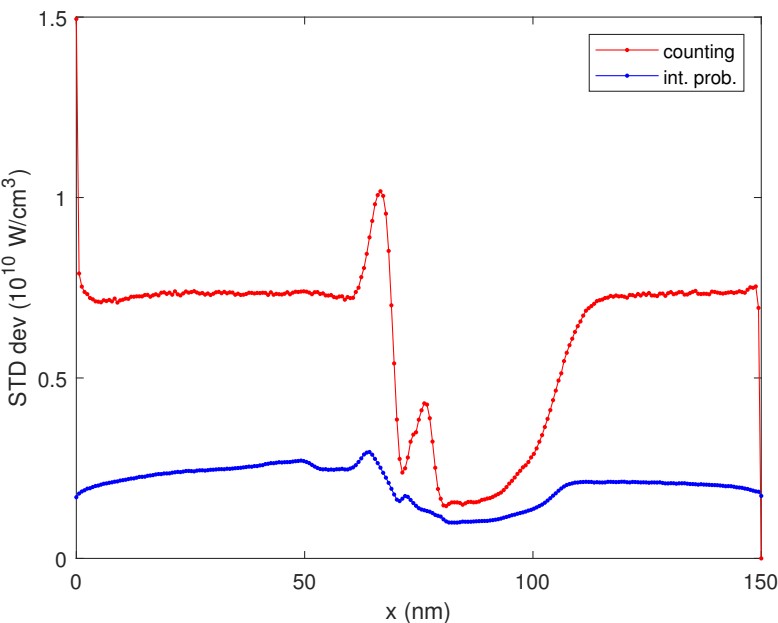

**Figure 3.** The standard deviation of the counting estimator (30) and the integrated probability estimator (32) versus position, for $V_b = 0.8$ V.

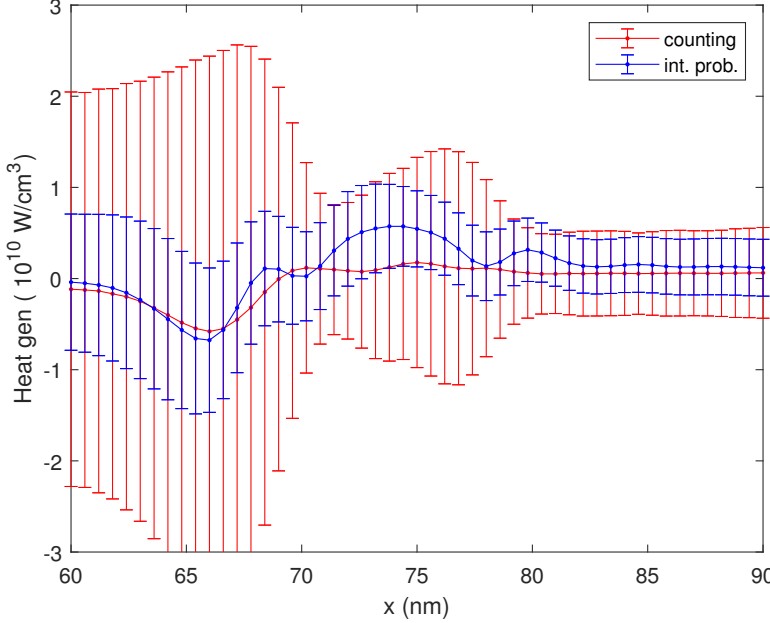

**Figure 4.** The heat generation rate vs. position for $V_b = 0.8$ V evaluated by means of the counting estimator (30) and the integrated probability estimator (32), with error bar.

Figure 6 shows the lattice temperature evaluated by means of the heat diffusion Equation (28) for the first two iterations, which are enough to reach the convergence. We observe that the lattice temperature is decreasing with the iteration number. To explain this behavior, one must consider the function $G(\varepsilon)$ in Equation (31). This function represents the difference between the emitted and absorbed phonon probability; if this quantity is positive,

more phonons are released into the lattice and in turn the temperature increases. We plot this quantity in Figure 7 showing that, for this particular kind of scattering mechanism, it decreases with the lattice temperature. In Figure 8, we plot the current versus the iteration number, proving that this quantity is constant. If we double the applied voltage to $V_b = 1.6$ V, the increase of temperature is of a factor 5 as shown in Figure 9.

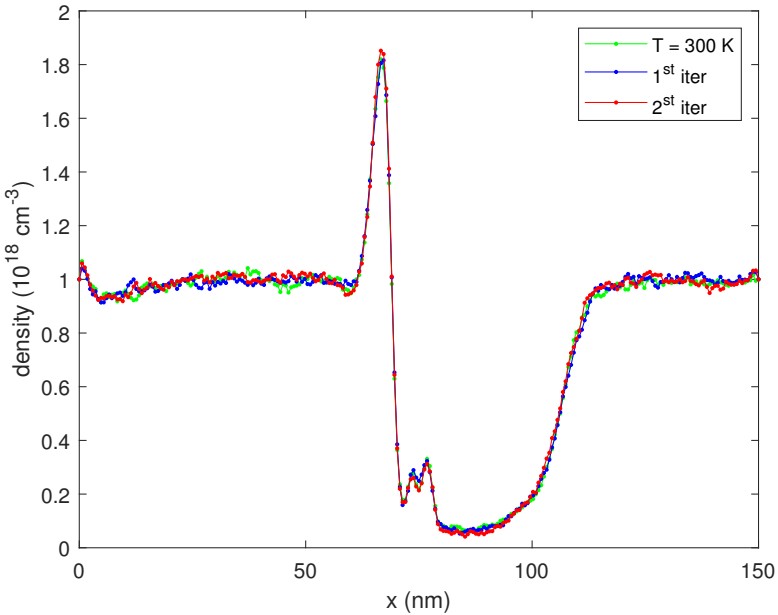

**Figure 5.** The density versus position for some iterations, for $V_b = 0.8$ V.

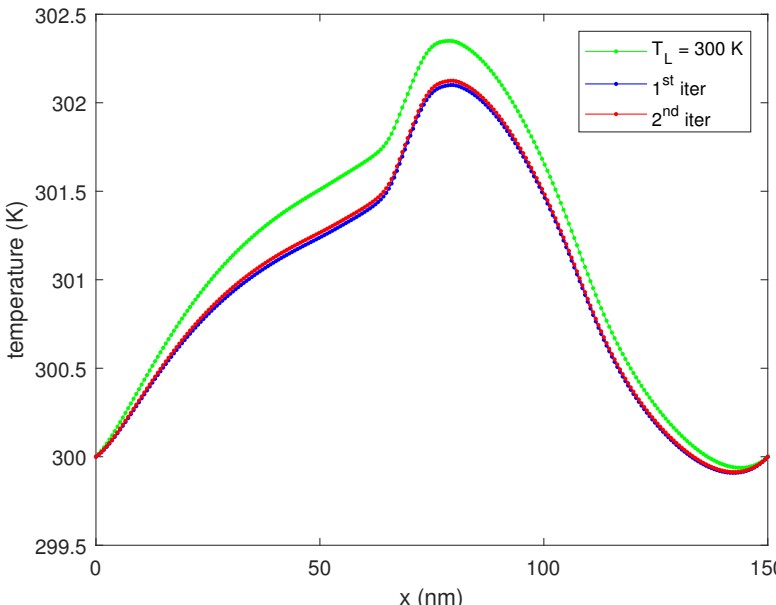

**Figure 6.** The lattice temperature $T_L$ versus position for some iterations, for $V_b = 0.8$ V.

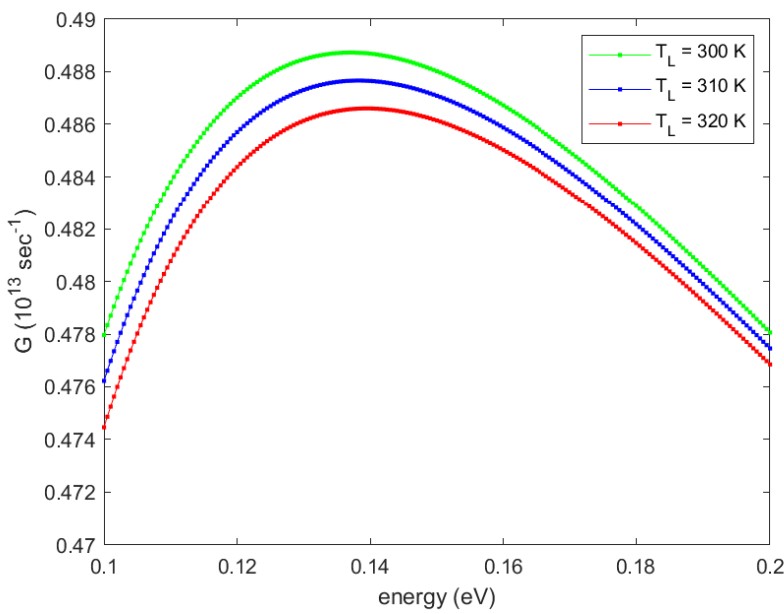

**Figure 7.** The function $G(\varepsilon)$ (31) versus energy for some lattice temperature.

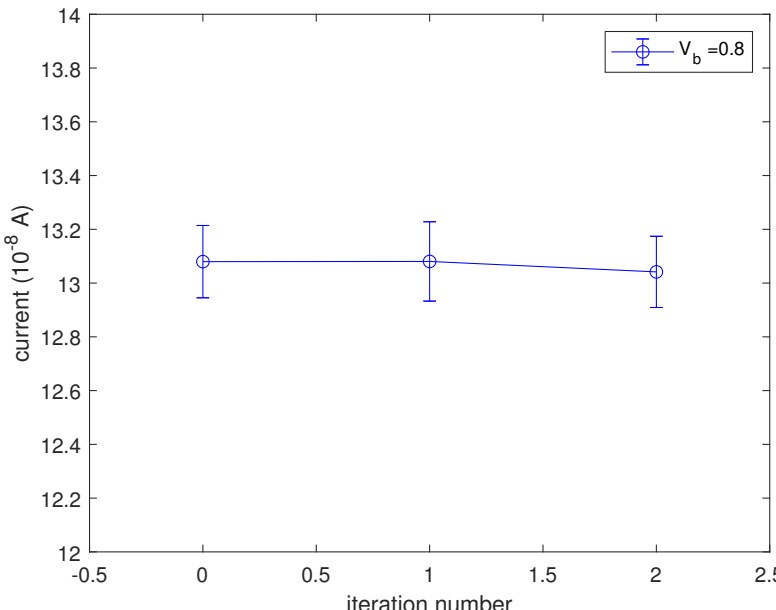

**Figure 8.** The current versus iteration number, for $V_b = 0.8$ V.

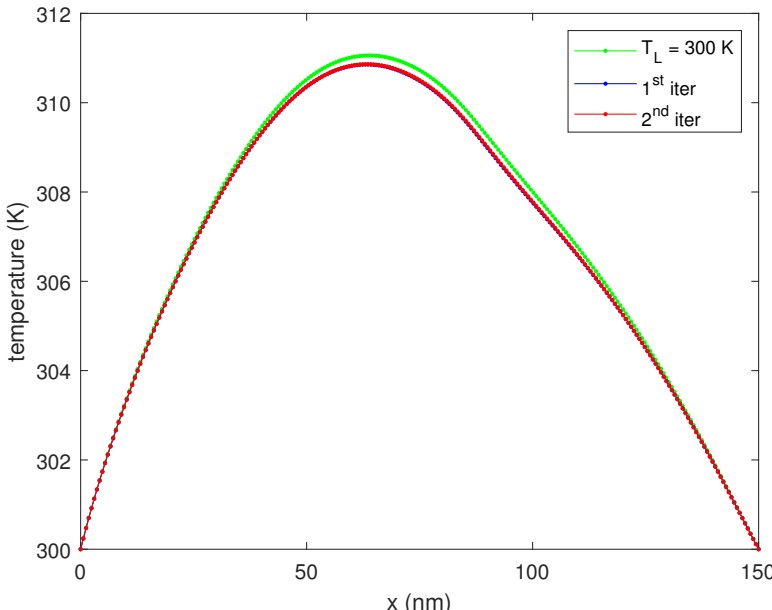

**Figure 9.** The lattice temperature $T_L$ versus position for some iterations, for $V_b = 1.6$ V.

## 7. Conclusions

The Electrothermal Signed Particle Monte Carlo algorithm provides an accurate tool for studying heat generation and quantum effects in nanometric semiconductor devices, at the expense of huge computational effort. The coupling between the MC charge transport and the heat diffusion equation is given by a term called heat generation rate obtained, usually, by counting the number of phonons emitted/absorbed during the steady-state. Alternatively, a new estimator of the heat generation rate, based on the integrated scattering probability function (32), can be used, which enjoys reduced statistical fluctuations. Simulation results for a Resonant Tunneling Diode are shown, proving that the heat is produced almost entirely inside the quantum well and estimating the lattice temperature, which depends on the applied voltage. The localization of hot spot regions can be useful in the design of such devices, in order to optimize the heat removal.

**Funding:** This research received no external funding.

**Data Availability Statement:** Data is contained within the article.

**Acknowledgments:** The author acknowledges the support of the project "Developing a Computational Framework for Quantum Information and Communication Technologies", Piano di incentivi per la ricerca di Ateneo 2020/2022—linea 2, Università degli Studi di Catania, and by the National Group of Mathematical Physics (GNFM-INDAM).

**Conflicts of Interest:** The author declares no conflict of interest.

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
