# Peer review of "Electrothermal Monte Carlo Simulation of a GaAs Resonant Tunneling Diode"

_axioms, doi:10.3390/axioms12020216_

Round 1

Reviewer 1 Report

I recommend to accept the paper as it is, however the authors can make it even better by taking into account the following comments:

1. The sentence on line 99 'In order to contain the particle number..' must be corrected, maybe  to 'In order to control...'.

2. The density n(x,t) defined in eqn 3 is later denoted by n(t,x), eqn 30 and 32, please make consistent.

3. Equations 30 and 32 provide the coupling between charge and heat transport and need more comments in order to make the work self-contained (without the need to read the given reference 25). For example the main term in eqn 30, the second ratio, has a  transparent physical meaning of  energy ballance between emitted and absorbed quanta of phonon energy. However the prefactor n/N_p is a ratio between a physical quantity and a numerical one ( related to the signed particle concepts) and thus needs explanation.

Author Response

"Please see the attachment

Reviewer 2 Report

This paper the electron transport and heat generation in a Resonant Tunneling Diode semiconductor device is studied. An electrothermal Monte Carlo method is presented. The method presented couples a Monte Carlo solver of the Boltzmann-Wigner transport equation with a steady-state solution of the heat diffusion equation. A microscopic description of the spatial distribution of self-heating and its effect on the detailed nonequilibrium carrier dynamics is provided.

The coupling between the MC charge transport and the heat diffusion equation is given by a heat generation rate.

Simulation results for a Resonant Tunneling Diode are shown, proving that the heat is produced inside the quantum well and estimating the lattice temperature which depends on the applied voltage. The results obtained can be useful in the design of devices, in order to optimize the heat removal. 

The text of the manuscript should be checked carefully again since there are some misprints (for instance, at the end of equations (1), (2), (4), (5), (6), (11), (17), (25), (30), and before the word “where” a comma must be inserted), etc.

Otherwise, the paper contains original results and can be published after minor amendments.

The presentation of the graphs and the tables are satisfactory.

The study performed by the author is relevant, interesting and, in general, the effort made is really appreciated.

Final recommendation: Publish after minor revision.

Author Response

The paper has been checked and misprints corrected.